# Carbon clusters formed from shocked benzene

D. M. Dattelbaum [1✉], E. B. Watkins[2], M. A. Firestone[2], R. C. Huber[1], R. L. Gustavsen[1], B. S. Ringstrand[2], J. D. Coe[3], D. Podlesak[4], A. E. Gleason [5], H. J. Lee[6], E. Galtier[6] & R. L. Sandberg[2,7]

Benzene ($C_6H_6$), while stable under ambient conditions, can become chemically reactive at high pressures and temperatures, such as under shock loading conditions. Here, we report in situ x-ray diffraction and small angle x-ray scattering measurements of liquid benzene shocked to 55 GPa, capturing the morphology and crystalline structure of the shock-driven reaction products at nanosecond timescales. The shock-driven chemical reactions in benzene observed using coherent XFEL x-rays were a complex mixture of products composed of carbon and hydrocarbon allotropes. In contrast to the conventional description of diamond, methane and hydrogen formation, our present results indicate that benzene's shock-driven reaction products consist of layered sheet-like hydrocarbon structures and nanosized carbon clusters with mixed $sp^2$-$sp^3$ hybridized bonding. Implications of these findings range from guiding shock synthesis of novel compounds to the fundamentals of carbon transport in planetary physics.

[1] Shock and Detonation Physics, Los Alamos National Laboratory, Los Alamos, NM, USA. [2] Materials Physics and Applications Division, Los Alamos National Laboratory, Los Alamos, NM, USA. [3] Theoretical Division, Los Alamos National Laboratory, Los Alamos, NM, USA. [4] Chemistry Division, Los Alamos National Laboratory, Los Alamos, NM, USA. [5] Fundamental Physics Directorate, SLAC National Accelerator Laboratory, Menlo Park, CA, USA. [6] Linac Coherent Light Source, SLAC National Accelerator Laboratory, Menlo Park, CA, USA. [7] Present address: Department of Physics and Astronomy, Brigham Young University, N261 Eyring Science Center, Provo, UT, USA. ✉email: danadat@lanl.gov

Astronomical observations have shown that carbonaceous compounds, such as refractory or icy solids, are ubiquitous in our galaxy and the cosmos. Understanding the creation of organic molecules and prebiotic material, and their voyage from the interstellar medium to the early solar system provides important constraints on the emergence of life on Earth. Impacts of remnant planetesimals (comets and asteroids) and associated shock-driven chemistry may have played a key role in the Origin of Life. However, understanding of such shockwave-driven chemical reactions of organic molecules is limited, largely due to the optically opaque, and complex product mixtures that form from nanoseconds (ns) to microseconds (μs) behind the shock front[1–6]. Very little is known of the initial reaction steps, reaction intermediates, or late-time product compositions, especially in real-time under shock compression conditions.

The π-stacked structure in Benzene-I (solid)[7–9], and $6p_z$ spatially extended electron density distribution make benzene susceptible to solid–solid phase transformations and chemical transformations upon compression[10,11]. Under static compression, benzene has a limited region of liquid phase stability and solidifies into at least three solid phases[9–11]. Recently, a theoretical investigation (DFT, VASP) found Benzene-V to be the most stable structure at $P > 40$ GPa, polymerizing into a Polymer-I structure above 80 GPa that is characterized by bridged-$C_6H_6$ layers[12]. Theory and modeling showed that at all high pressures examined, saturated, four-coordinate ring-containing structures, (i.e., graphanes) were more stable than benzene[12]. It was proposed by Engelke et al., in a theoretical study, that compression drives overlap of π-electron densities leading to dimerization or polymerization[13].

Shock-driven reactions often occur at lower pressures compared with static high-pressure reactions due to the high temperatures reached along the shock adiabat. Moreover, shock-driven products formed from liquid reactants differ from solids due to the lack of molecular ordering which promotes addition reactions within a unit cell. States along the principal shock locus (Hugoniot) for benzene have been reported by several groups[2,14–18]. Under shock loading benzene has been shown to exhibit a "cusp" on the shock adiabat near 13 GPa[2,14–16], which can be seen in the pressure-specific volume (P–V plane) in Fig. 1 (inset) as a deviation to reduced volume in the Hugoniot data from Dick[19], Nellis[16], Lysne[20], and Dattelbaum[14]. Near 20 GPa, the transformation of benzene to products occurs over ~180 ns, with a total volume decrease of 12.5%[14].

There have been several attempts at measuring the shock-driven chemistry of benzene using optical techniques[1,15,21–23]. Yakusheva et al. observed optical absorbance changes 300 ns behind the shock front above ~13.5 GPa[24]. The shock-driven opacity was found to be irreversible, and it was speculated that the yellow color resulted from light scattering off carbon particles. Holmes et al. proposed that carbon particles that preferentially absorb in the blue region form behind the shock front on similar timescales[25]. Root and Gupta concluded that benzene remained in the liquid phase under quasi-isentropic loading to 13 GPa but at lower temperatures than the principal Hugoniot, suggesting that a liquid–solid transition in benzene does not occur on shockwave timescales[21,22]. More recently, Bowlan et al. found that benzene was unreactive when shocked to 18 GPa for 300 ps[1]. However, to date, there have been no in situ experimental confirmations of the composition of benzene's shock-driven reaction products.

Theoretical insights into shock-driven reactions in benzene have also been pursued. For example, Maillet and Pineau[26], using the ReaxFF reactive force field, found that at simulated shock pressures just above the cusp, C–C bonds formed into aliphatic chains as benzene underwent ring-opening reactions with adjacent molecules. As pressure (P) and temperature (T) increased to conditions similar to those studied here ($V = 0.416$ cm$^3$/g, $T = 4450$ K), $H_2$ was found to form, and a significant percentage of C–H bonds were replaced by C–C bonds. More recently, Martinez et al.[27] performed simulated shock experiments at several pressures using extended Lagrangian Born–Oppenheimer molecular dynamics with parallel replica dynamics and found that above 18 GPa benzene dimerized via several pathways including H–H elimination from end-on condensation; a precursor to polymerization.

Here, we investigate the shock-driven chemistry of benzene, a central molecule in organic chemistry both as a building block for molecular compounds, and as a model of chemical stability derived from its $4n + 2$ π-electron cyclic aromaticity, using in situ femtosecond (fs)-duration x-ray pulses from an x-ray free-electron laser (XFEL) at the Linac Coherent Light Source (LCLS)[28–30]. High brilliance x-rays have only recently been used to probe transformations in solid carbon under shock wave compression and detonation[31–36]. We report the transformation of liquid benzene to solid products on the timescale of the shock duration (10–20 ns) and identify carbon or hydrocarbon product species formed at these extreme conditions through analysis of x-ray diffraction and small-angle x-ray scattering. These carbon and hydrocarbon forms were interpreted as intermediate states in the transition from graphite to diamond forms.

## Results

**Shock-loading conditions.** Liquid benzene ($C_6H_6$, $\rho_0 = 0.876$ g/cm$^3$) at room temperature was shocked to 27 ($\pm 4$) and 55 ($\pm 5$) GPa in five separate experiments by laser-driven shock compression at the matter in extreme conditions (MEC) endstation at LCLS. Under these shock input conditions, benzene is rapidly ($\tau < 1$ ns, $k \sim 10^3$ μs$^{-1}$) converted to shock-driven reaction products[14].

Equations-of-state (EOS) for liquid benzene and its shock-driven decomposition products have been reported previously[14–16], and new EOS were developed for the present work. Details are provided in the "Methods" section and Supplemental Information, but here we note that their primary purpose was to provide estimates of shock temperature for the kinetically trapped, non-equilibrium states observed in the experiments. Because EOS frameworks are almost universally based on the assumption of full thermodynamic equilibrium, we first built equilibrium reactants and products EOS that agreed with Rankine–Hugoniot observables in their respective domains on the principal Hugoniot. We then estimated experimental temperature at a given pressure as the mean of those of reactant and products, with an uncertainty equal to half their difference; this procedure was intentionally meant to yield conservative estimates of uncertainty. Comparison of the EOS results to Rankine–Hugoniot data are shown in the inset of Fig. 1, and approximate shock temperatures are reported in Table 1.

The shock-driven transformation of liquid benzene to solid products was evaluated under sustained (10–20 ns) shock compression using both x-ray diffraction (XRD) and small-angle x-ray scattering (SAXS) of monochromatic x-rays. XRD elucidates the crystal structures formed during the dynamic drive, while SAXS provides a means to evaluate nanoscale product morphology[37].

**Small-angle x-ray scattering.** Analogous to ns-time-resolved SAXS studies of the temporal evolution of carbon formed from high explosive detonation, the SAXS signal from shocked benzene products likely originates from electron density contrast between solid products and a fluid matrix[33–35,38]. Shown in Fig. 2 are 1-D SAXS profiles of benzene at an input pressure of $55 \pm 5$ GPa (panel A) and at $27 \pm 4$ GPa (panel B). A qualitative

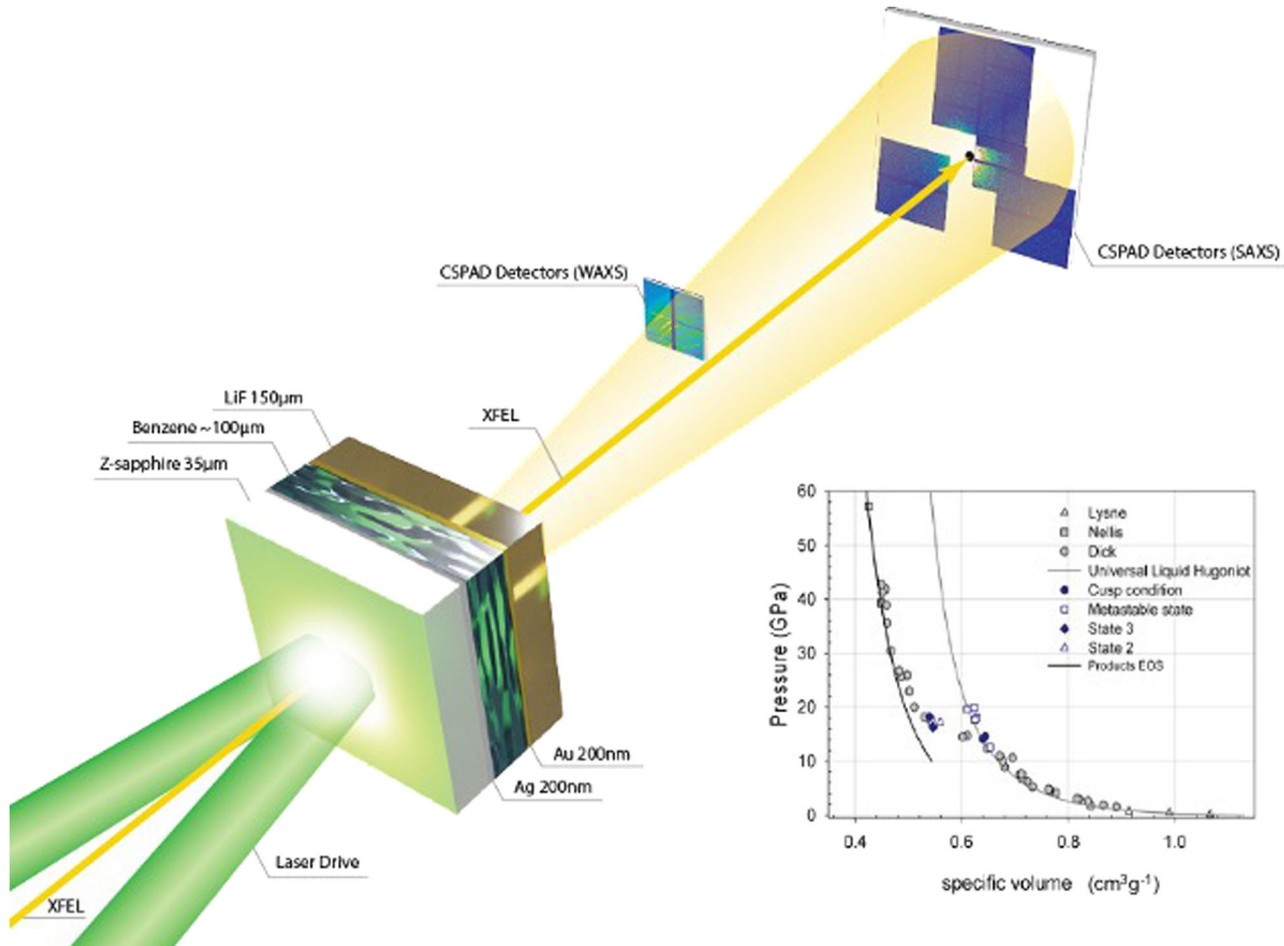

**Fig. 1 Experimental configuration of laser-driven shock compression experiments on benzene.** A 100 μm-thick droplet of benzene was placed between an ablator window of coated z-cut sapphire and a rear single crystal [100] LiF window and held in place with a Teflon o-ring. XRD and SAXS were collected using a series of seven detectors place at a range of angles and distances from the target (see the "Methods" section). For clarity, not all of the seven detectors are shown. Shock states in liquid benzene from several literature sources (Dick[19], Nellis[16], Lysne[20], and Dattelbaum[14]) obtained using traditional gas gun-driven plate impact techniques are shown in the pressure–volume plane in the plot. A "cusp" or deviation in the Hugoniot in the P–V plane is observed at 13.8 GPa and is purported to be due to a shock-driven chemical reaction with a volume change of −12.5%.

examination of the 55 GPa data reveals a primarily power-law dependence of the scattering, indicating that the length scales associated with the solid products were larger than the measurement window ($R_g > 15$ nm). The power-law ($P_1$) was, with the exception of run 239, about −4 indicative of smooth-surfaced product morphologies. Thick solid lines are fits to a 2-level model consisting of a power-law ($P_1$, thin solid lines) representing larger length scales, a minor contribution from a Guinier–Porod level ($G_2$, $R_{g2}$, $P_2$, dashed lines) potentially indicating a ~1 nm length scale in the products ($R_{g2} \sim 5$–15 Å), and a constant background (dotted lines) (Fig. 2). In the case of run 303, the addition of the Guinier–Porod level did not significantly improve the fit. Scattering from the products formed at a shock pressure of 27 GPa (run 292, panel B) was significantly different, exhibiting a slight bump at ~0.04 Å$^{-1}$, and could not be well approximated by a simple power-law form. Combining a low-$q$ power-law with a Guinier–Porod contribution was capable of reproducing the features of the SAXS data, indicating a ~4 nm length scale associated with the reaction products. Alternatively, the data could be modeled without a low-$q$ power law by using a log-normal distribution of Guinier–Porod contributions. Here, we correlated the width of the distribution to the distribution mean based on kinetic coagulation models[39]. This approach yielded a comparable product length scale of 5 nm.

**Wide-angle x-ray diffraction.** Insight into the identity of the nanoparticles was obtained through analysis of the XRD data. Following shockwave compression, intense diffraction lines emerge (Fig. 3C red), with strong texture (azimuthal intensity variations) captured on the 2D detector images (Fig. 3B). In control measurements of shocked LiF/Sapphire assemblies performed between each experiment, none of these peaks were observed; they are attributed to diffraction from benzene reaction products.

Integrated diffraction patterns were analyzed to identify the structure of the solid products (Fig. 4A). All of the XRD patterns show strong reflections in the scattering vector range spanning 1.8–1.9 Å$^{-1}$ (Table 1), roughly in the region of the (002) reflection corresponding to the interlayer spacing of graphite, disordered graphitic carbons (e.g. carbon black, glassy carbon) and graphane polymorphs (Fig. 4B)[40]. A single diffraction peak located at $q = 1.88$ Å$^{-1}$ (d-spacing of 3.34 Å) is observed for run 237 (Fig. 4B, blue), run 239 (Fig. 4B, red) contains a single peak at $q = 1.80$ Å$^{-1}$ (d-spacing = 3.49 Å), and two reflections ($q = 1.81$ Å$^{-1}$; d-spacing = 3.47 Å and $q = 1.88$ Å$^{-1}$; $d = 3.34$ Å) are observed for run 303 (Fig. 4B, green). The observed multiplicity of the low-$q$ peak and/or variation in peak position from run-to-run can be attributed to differences in the layer spacing[40–42]. For comparison, at ambient conditions h-graphite

**Table 1 Summary of shock conditions, and SAXS analysis of the benzene reaction products.**

| Exp. # | P (GPa) | T (K) | d spacing [q (Å⁻¹)] | $B_1$ | $P_1$ | $G_2$ | $R_{g2}$ (Å) | $P_2$ | $\chi^2_{1,2}$ | $\chi^2_1$ |
|---|---|---|---|---|---|---|---|---|---|---|
| 237 | 55 ± 5[a] | 4940 ± 710 | 3.4 Å [1.850] | 0.009 ± 0.004 | 4.02 ± 0.14 | 16.3 ± 10.2 | 9.0[b] −2.8, +1.6 | 4[c] | 2.3 | 5.2 |
| 239 | 55 ± 5 | 4940 ± 710 | 3.6 Å [1.766] | 0.021 ± 0.006 | 3.46 ± 0.09 | 4.4[b] −2.6, +4.4 | 8.1 ± 3.6 | 4[c] | 0.9 | 1.2 |
| 292 | 27 ± 4 | 2790 ± 455 | 3.4 Å [1.80] | 0.005 ± 0.004 | 3.94 ± 0.32 | 1390[b] −1110, +3890 | 38.2[b] −8.8, +15.7 | 4[c] | 4.1 | 5.4 |
| 294 | 55 ± 5 | 4940 ± 710 | 3.4 Å [1.85] | 0.015 ± 0.006 | 4.14 ± 0.21 | 56.2[b] −39.4, +70.3 | 12.1 ± 2.3 | 4[c] | 5.6 | 7.2 |
| 303 | 55 ± 5 | 4940 ± 710 | 3.4 Å [1.85] | 0.019 ± 0.016 | 3.85 ± 0.37 | — | — | — | 7.4 | 7.7 |

Parameters of a 2-level unified model are presented ($R_g$ is the radius of gyration, $G$ is the Guinier scaling factor, $B$ is the Porod scaling factor, and $P$ is the Porod exponent). The goodness-of-fits ($\chi^2_{1,2}$) are compared to best fits using just a power-law and constant background ($\chi^2_1$). Also listed is the layer spacing of the products obtained from the low-angle XRD reflection. Estimated temperatures are the mean of those of reactant and products, with an uncertainty equal to half their difference, as described in Supplementary Note 1.
[a]The remaining shock parameters are, for the reactant: $P = 27$ GPa, $U_s = 7.8$ km/s, $u_p = 4.0$ km/s, and at $P = 55$ GPa, $U_s = 10.4$ km/s, $u_p = 6.0$ km/s. On the product Hugoniot at $P = 27$ GPa, $U_s = 7.3$ km/s, $u_p = 4.2$ km/s, and at $P = 55$ GPa, $U_s = 10.0$ km/s, $u_p = 6.3$ km/s.
[b]Positive and negative parameter errors are given for asymmetric error ranges—a 2-level fit improved $\chi^2$ by <10%.
[c]Fixed parameters.

in the ABAB stacking arrangement has a $d$-spacing of 3.35 Å ($q = 1.88$ Å⁻¹), turbostratic graphite with rotationally disordered plane stacking has a $d$-spacing of 3.44 Å ($q = 1.83$ Å⁻¹), and highly curved and disordered carbon forms such as glassy carbon and carbon onions can have $d$-spacings as large as 3.8 Å ($q = 1.65$ Å⁻¹)[40,42]. However, these $d$-spacing do not account for the compression predicted for the high $P$–$T$ conditions measured here. While graphite is not stable at the measured $P$–$T$ conditions, extrapolation of the high P EOS of graphite yields compressed $d$-spacings for $h$-graphite to glassy carbon ranging from 2.7–3.1 Å at 27 GPa and 2.6–2.9 Å at 55 GPa[43]. Thermal expansion under these conditions is relatively small yielding approximately on Hugoniot $d$-spacings for $h$-graphite to glassy carbon ranging from 2.9 to 3.3 Å at 27 GPa, 2790 K, and 2.7–3.1 Å at 55 GPa, 4940 K[44]. The predicted high $P$–$T$ $d$-spacings for carbon interlayer spacings are all significantly larger than those measured indicating that graphite is not a major constituent of the reaction products and that disordered graphitic-like carbons with a reduced degree of planarity of the sheet are not sufficient to produce the expansion in the layer spacing that is observed[40–42]. Instead, the expanded layer spacing may be attributed to graphane forms composed of hydrogenated carbon sheets. Multilayer graphane also referred to as hydro-graphite or graphate, has been theoretically predicted and synthesized[45,46]. Experimentally, stable hydro-graphite adopts a graphate-II or "buckled" structure composed of weakly coupled single graphane layers in a chair conformation. Predicted structures of graphane I–IV at 50 GPa and ambient $T$ exhibit (002) diffraction peaks between 1.83 and 1.87 Å⁻¹, consistent with the low-$q$ peaks measured here. While the thermal expansion for the predicted graphane structures was not calculated, graphite exhibits a relatively small $d$-spacing shift at elevated $T$ which suggests the inclusion of thermal expansion will not significantly impact the interpretation of the results. The predicted structure of a mixed $sp^2$–$sp^3$ carbon phase, $H_{18}$, at 50 GPa and ambient $T$ exhibits a diffraction peak at 1.76 Å⁻¹, slightly below the $q$ position of the measured peaks[47]. Considering both the uncertainty associated with crystal structures of predicted phases and the small $d$-spacing shift anticipated due to thermal expansion, this may also represent a candidate structure in the benzene reaction products.

The full-width at half maximum (FWHM) of diffraction peaks is an established indicator of crystallite size (i.e., the number of stacked sheets in the registry) by the Scherrer equation[48]. The narrowest low-$q$ reflection is found for run 237 (FWHM = 0.037 Å⁻¹; $L_c$ ~ 150 Å) while the broadest peak is observed for run 239 (FWHM = 0.048 Å⁻¹; $L_c$ ~ 120 Å). Run 303, contains two resolvable peaks, suggesting the co-existence of varying multilayer populations (FWHM $q = 1.81$ Å⁻¹ = 0.044 Å⁻¹; $L_c$ ~ 130 Å; FWHM $q = 1.88$ Å⁻¹ = 0.095 Å⁻¹; $L_c$ ~ 60 Å). While these crystallite sizes are accessible by the SAXS measurements, we did not detect comparable length scales and the SAXS signal was dominated by larger structures suggesting that the solid products are internally disordered and composed of multiple domains.

In addition to the expanded $d$-spacing indicated by the low-$q$ XRD, several peaks in the higher $q$ regions of the XRD pattern could not be indexed to conventional carbon phases. For example, peaks in the 2.4–2.8 and 3.4–3.6 Å⁻¹ range do not correspond to graphite, disordered graphite-like, or diamond structures. In general, it was not possible to index all of the XRD peaks to known or predicted carbon or hydrocarbon structures indicative of a highly complex mixture of solid products. Thus, our approach was to make qualitative comparisons between the measured XRD pattern and diffraction from representative carbon and hydrocarbon structures capable of reproducing the measured low-$q$ diffraction peak, namely graphane forms

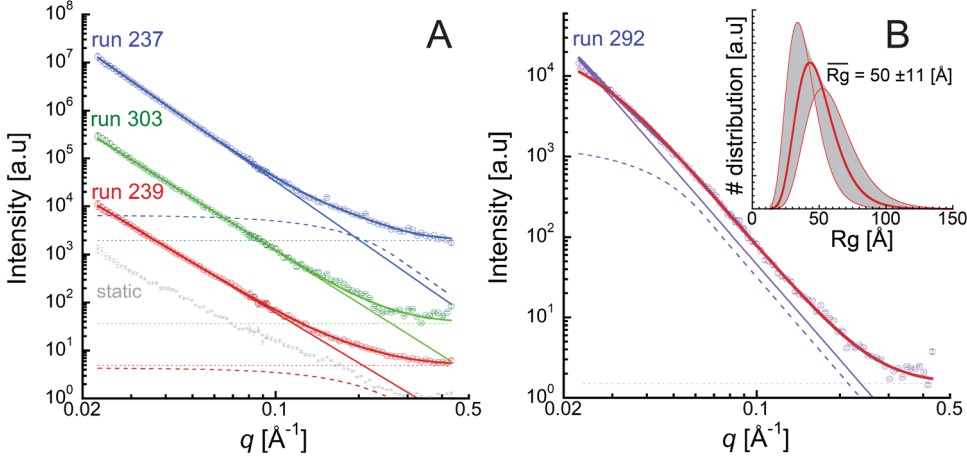

**Fig. 2 SAXS profiles from benzene products formed within 20 ns of shock input.** The SAXS profiles correspond to input conditions of $P = 55 \pm 5$ GPa, $T = 4940 \pm 710$ K (panel **A**) and $P = 27 \pm 4$ GPa, $T = 2790 \pm 455$ K (panel **B**). **A** SAXS data (symbols) after subtraction of the static intensity (representative static measurement shown in gray): run 239 (red), run 303 (green), and run 237 (blue). Runs 237 and 303 are offset vertically for clarity. Thick solid lines are fits primarily consisting of a power-law ($P_1$, thin solid lines) representing larger length scales and additional contributions from a Guinier–Porod level ($G_2$, $R_{g2}$, $P_2$, dashed lines) corresponding to a ~1 nm length scale, and a constant background (dotted lines). In the case of run 303, the addition of the Guinier–Porod level did not significantly improve the fit. **B** SAXS data (purple symbols) for lower pressure measurement (run 292), a 2-level model (purple lines), and intensity from Guinier-Porod levels with a log-normal $R_g$ distribution (red line). The inset shows the modeled log-normal distribution of product sizes, with a mean $R_g$ of 50 Å, and shaded ranges associated with the parameter errors.

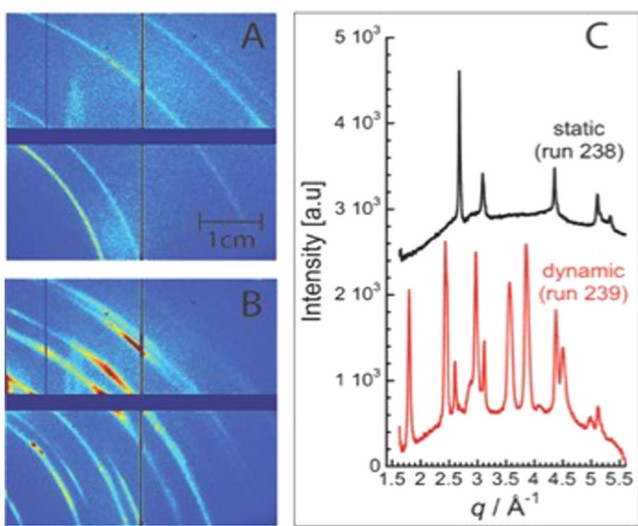

**Fig. 3 X-ray diffraction data is shown for benzene before and after shockwave compression.** Static (**A**) and dynamic (**B**) XRD patterns were collected on CSPAD detectors located $x = 6$ cm from the benzene sample. (color map corresponds to recorded intensity) Diffraction lines from polycrystalline Au coating are observed in the static image. **C** Upon shockwave compression to 55 GPa, diffraction lines are observed from crystalline products formed from shocked benzene. The integrated pattern (in arbitrary units) in (**C**) shows diffraction peaks at this condition ($P = 55 \pm 5$ GPa, $T = 4940 \pm 710$ K). The color scale is relative intensity and is arbitrary to the CSPAD detector signal.

and the mixed $sp^2$–$sp^3$ hybrid phase $H_{18}$. In all cases, the majority of the peaks do not correspond to a graphite or diamond structure but several have positions comparable to graphates or $H_{18}$ (Table S1–S3)[46,47]. While some reflections are consistent with cubic or hexagonal diamond, they cannot be unambiguously indexed and may also be attributed to graphate or $H_{18}$. The possibility of cubic diamond products agrees with the findings of Kraus et al. who reported that shock compression of both polycrystalline and pyrolytic graphite transitions to the diamond

at ~50 GPa[49]. However, Kraus reported coexisting diffraction signatures ($q \sim 2.2$ Å$^{-1}$) for a lower pressure (19 GPa) compressed graphite (002) layer spacing, whereas we observe diffraction peaks between 1.8 and 1.9 Å$^{-1}$ indicating an expanded layer structure. This suggests that the reaction products follow a significantly different transformation pathway than the graphite to diamond transition. Evidence for a mixture of both hydrogenated graphite and mixed $sp^2$–$sp^3$ hybrid phases may be interpreted as the shock-induced polymerization of benzene into layered hydrogenated carbon that is undergoing transformation into an $sp^3$ diamond-like material (Fig. 5B).

## Discussion

Carbon has been found to exist in a wide variety of complex allotropes, both experimentally and theoretically, due to its ability to readily form $sp$-, $sp^2$, and $sp^3$-bonds. The transition between $sp^2$ and $sp^3$ carbon forms has been found experimentally to occur under a variety of conditions[50–54]. Nano-onions, amorphous carbon and diamond have all been found in recovered carbon soot from detonated explosives, and recently measured in situ using time-resolved small-angle x-ray scattering[33,34,55].

Shock-driven chemical reactions are not expected to be simple due to the instantaneous nature of uniaxial shockwave compression, and often a mixture of products is found in recovered samples. Previously, under single shock compression, highly oriented graphite was found to produce cubic diamond at pressures >20 GPa and a hexagonal form of diamond (lonsdaleite) above 170 GPa[56]. Here, we do not see compelling evidence of either cubic or hexagonal diamond as dominant constituents of the solid products of benzene. From combined in situ x-ray diffraction and small-angle scattering, a description of the transformation of benzene emerges that is consistent with intermolecular polymerization into hydrogenated graphite-like sheets that undergo densification to $sp^3$-carbon on the pathway to diamond (Fig. 5B).

The states reached in shocked benzene are within the phase-stability regions were hexagonal and cubic diamond, and liquid carbon phases might be expected to exist (Fig. 5A). Yet, the observed experimental variations in carbon allotropes are

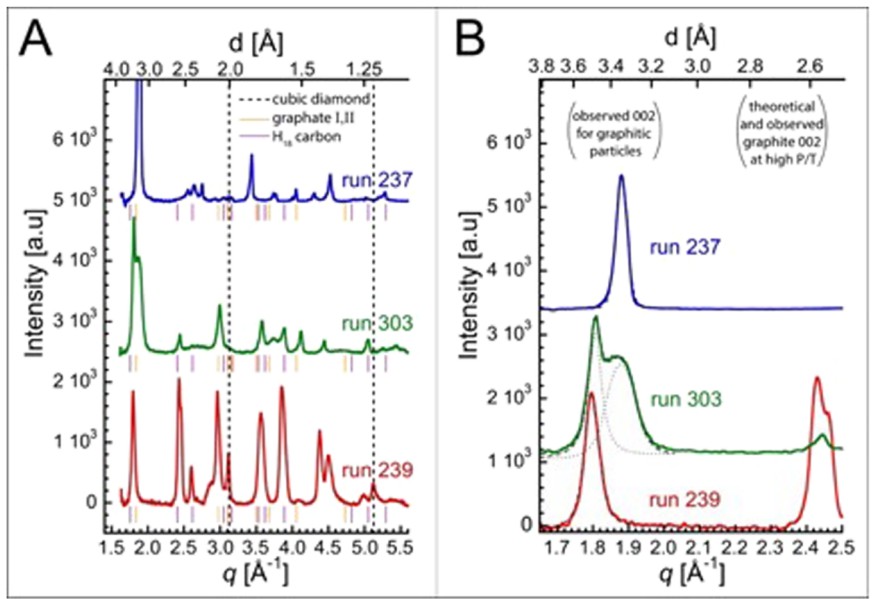

**Fig. 4 Diffraction patterns (in arbitrary units) of solid products formed from shocked benzene at 55 GPa, 4940 K. A** Expanded region between $d = 2.5$ and 3.8 Å shows the measured low-angle peaks (solid lines) and fits (dashed lines) relative to the estimated range of $d$-spacings for graphitic layered carbon structures at the same $P$–$T$ conditions (gray shaded region) and the range of predicted d-spacings for graphate I–IV and $H_{18}$ at 50 GPa and ambient $T$ (red shaded region). **B** Full range of the measured diffraction exhibiting multiple peaks that do not index to graphite or diamond forms, particularly in the regions of 2.4–2.8 and 3.4–3.6 Å$^{-1}$. Dotted lines correspond to estimated graphite reflections, dashed lines to cubic diamond reflections at 55 GPa and 5000 K. Calculated diffraction patterns for graphate-I, and -II from Wen et al. [46], and $H_{18}$ calculated at 50 GPa and ambient $T$ are shown for comparison[47].

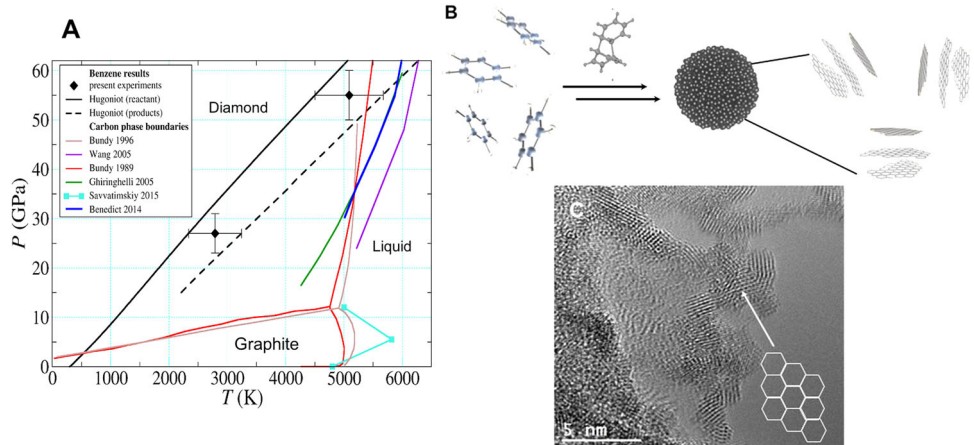

**Fig. 5 The products of shock-driven benzene on the principal Hugoniot form from dimerization and polymerization, followed by condensation into clusters with the sheet-like structure of $sp^2$–$sp^3$ character. A** Carbon phase diagram showing diamond, graphite and liquid regions from several works (see legend)[62–64,66,71,72]. A region of transition from $sp^2$–$sp^3$ hybridized allotropes is proposed to occur in the diamond region by Blank et al. [73–75]. $P$–$T$ states along the principal reactants and products Hugoniots, and the calculated $P$–$T$ states from shocked benzene (this work) are overlaid on the phase diagram (errors as reported in Table 1 and Supplementary Note 1). **B** Schematic of the proposed mechanism of cluster formation from benzene. Shocked liquid benzene undergoes dimerization and polymerization addition reactions to form clusters with disordered, hydrogenated layered carbon structures with $sp^2$–$sp^3$ character. **C** Transmission electron micrograph of recovered carbon from PBX 9502, and explosive that samples a similar $P$–$T$ condition to the experiments described here, showing the transition from layered-$sp^2$ to $sp^3$-diamond-like structures within the structure. Recovery of products from the laser-driven shock compression experiments of benzene was not feasible.

suggestive of kinetic trapping of non-equilibrium states at the studied shock conditions. Non-equilibrium product compositions have been previously suggested by measurements by Nellis et al.[16]. A key difference between shock-driven reactions in benzene and shock-driven solid–solid phase transformations in solid carbon is the compression energy, and thus the temperature rise imparted to the material on the shock adiabat. The compression energy in benzene shocked to 55 GPa ($E = 1/2 (P_1 - P_0)$ $(V_0 - V) = 26.5$ GPa cm$^3$/g) is far greater than that for graphite

shocked to the same shock pressure ($E = 5$ GPa cm$^3$/g)[57,58]. The increased compression energy results in higher temperature and greater entropy rise on the principal Hugoniot at the same shock pressure[36,49,57,58]. We expect these effects, and starting from a disordered liquid phase, drive the products to more-disordered solid structures, and result in incomplete conversion to the diamond at the measured shock conditions.

There is precedent for layered carbon structures produced from both explosive detonation and planetary conditions. Recent high-

resolution transmission electron micrographs of nanophase carbons recovered from open-air detonations of a liquid explosive mixture of nitromethane 95 (v/v)% and diethylenetriamine 5 (v/v)% sensitizer showed graphite-like stacks (layering in the 002 direction) composed of ~7 to 10 atomic layers with an average interplanar distance of ~3.5 Å and intraplanar undulations (i.e., "wrinkles") in the carbon sheets[59]. Shungite rocks (group 5 deposits), a naturally occurring allotrope of carbon, are made up of globules (~6 nm) of imperfectly packed graphene clusters (5–6 layers) with a (002) reflection at $q = 1.77$ Å$^{-1}$ (d-spacing of 3.55 Å)[60,61]. The strong low-$q$ reflection observed in all the experiments reported here is also a hallmark of a layered structure. However, here the layer-to-layer thickness is expanded relative to high-pressure graphite, as noted by the diffraction peaks between 1.8 and 1.9 Å$^{-1}$, and is likely due to hydrogenation of the graphite sheets. An observed multiplicity in the peak and/ or variation in peak position from run-to-run can be attributed to differences in the layer spacing potentially representing different states along the transformation pathway[40].

The location of several higher-$q$ XRD reflections compare favorably with either hydrogenated graphite or a mixed $sp^2$–$sp^3$ hybridized bonding network, known as $H_{18}$, proposed to exist in detonation products[47]. The presence of a mixed $sp^2$–$sp^3$ hybridized species in several experiments is consistent with graphite transitioning to diamond, and may suggest it is an important intermediate in the condensation process. The existence of a mixed $sp^2$–$sp^3$ network is also consistent with recent findings from pressure-shear diamond anvil cell experiments by Blank et al. [62–64], in which a region of nano-onion stability in the carbon phase diagram with layered $sp^2$–$sp^3$ bonding motifs was observed at pressures between 50 and 100 GPa.

The early stages of reaction in benzene, with dimerization leading to polymerization, have been predicted computationally by several groups[12,26,27]. At simulated shock pressures just above the reaction cusp, C–C bonds formed into aliphatic chains as benzene underwent ring-opening reactions with adjacent molecules. As $P$ and $T$ were increased in the simulations to conditions similar to those studied here ($V = 0.416$ cm$^3$/g, $T = 4450$ K), H$_2$ was found to segregate out, and a significant percentage of C–H bonds were replaced by C–C bonds. Recent theoretical studies of 3-D graphane crystal structures by Wen et al.[46] calculated the relative enthalpies of different graphane polymorphs as a function of pressure. From the calculations, the chair conformers Graphane I (chair1 AA stacking) and II (chair 1 AB stacking) were found to be most stable at ambient pressure, consistent with the stability of isolated chair versus boat cyclohexane rings. Compression is expected to increase the stability of Graphane III (a distorted chair 2 structure) above 20 GPa, and Graphane IV (boat 1), a boat structure, becomes favored over chair I and II structures above 50 GPa. $H_{18}$, the quasi-layered mixed $sp^2$–$sp^3$ layered structure with bridging C–C bonds between the layers, has a density ($\rho_0 \sim 3.2$ g/cm$^3$ at 23 °C) that is intermediate to graphite and diamond, and has a bulk modulus (360 GPa) similar to diamond due to the $sp^3$-interlayer bridges. Analysis of the diffraction patterns from runs 237, 239, and 303 show reflections consistent with several graphate forms and $H_{18}$, indicating we have observed the transition of benzene into polymerized, hydrogenated sheets, with at least some degree of $sp^3$-interlayer bonds.

In summary, we have observed crystalline solid products from shock-compressed benzene using combined XFEL-SAXS and XRD. The solid product composition initially formed is complex, and indicates that neither diamond or graphite is the major constituent, illustrating the limitation of thermochemical equilibrium approaches to modeling of decomposition products at time scales <10 s of ns due to slow carbon kinetics[14,16]. Additional

experiments across a range of shock input conditions are necessary to further investigate the influence of the thermodynamic state on the product composition and non-equilibrium phases. In situ XFEL-based x-ray scattering and diffraction have illustrated the complexity of solid carbon and hydrocarbon products formed from simple molecular species in extreme conditions.

## Methods

**Materials.** Benzene was purchased from Sigma Aldrich (99.999%), and used without further purification. Liquid samples were pipetted into the sample volume of a gasketed cell comprised of the coated ablator, and rear window, and sealed using a liquid cell design used previously[65]. A cartridge design was used to load multiple liquid samples and positioned at the laser interaction position within the target chamber.

**Shock wave experiments at LCLS.** Shockwave compression experiments were performed at the MEC endstation at the LCLS. A 20 J, 5–20 ns-duration long-pulse compression laser was overlapped spatially and temporally with x-ray bunches from the LCLS XFEL. A 250 μm diameter laser spot generated by phase plates on the drive laser (Nd:Glass, $\lambda = 527$ nm) was used to create a flat-topped shock wave within the benzene via a laser ablation process within the Al-coated z-cut sapphire (Fig. 1)[66]. The polished z-cut sapphire wafers were coated with 200 nm of Al and 150 nm of Au using magnetron sputtering. The x-ray probe from the XFEL was ~50 fs duration and 25 μm in diameter with $E = 11$ keV, and an average of ~$10^{12}$ photons per pulse. A line-imaging velocimetry interferometer for any reflector (VISAR) was used to record the spatial characteristics and particle velocity of the transmitted shock at the benzene–LiF window interface using a 200 nm Al coating.

Laser power in individual experiments was chosen to overdrive the reaction transition in benzene via an ablation-driven shock generation process as described elsewhere. Doing so resulted in a single wave condition with prompt chemical reaction rate(s) $O$ (<1 ns), or less than the rise time of the shockwave. Furthermore, the experiments were timed such that the shockwave had traveled ~90% of the distance through the benzene sample when the x-ray measurement was made (timing jitter of 30–50 ps). This was done to ensure that states on the principal (not second shock) Hugoniot were probed. Shock states were determined using standard impedance matching techniques using the measured interface particle velocity to the LiF Hugoniot ($\rho_0 = 2.638$ g/cm$^3$, $s = 1.35$, $c_0 = 5.15$ mm/μs), as well as verified using an ablator-window only experiment at the same laser power to measure the shock input independently. It was estimated that the input pressure varied by <3% shot-to-shot at the same laser power.

X-ray diffraction (XRD) and small-angle x-ray scattering (SAXS) were collected using a total of seven detectors that were placed at various distances from the sample within and external to the MEC target chamber. Two Cornell-SLAC pixel array detectors (CSPADs) (sample-to-detector distance $x \sim 6$ cm) were used within the chamber for x-ray diffraction measurements from $6.0 > q > 1.6$ Å$^{-1}$. Azimuthal coverage of the detectors varied as a function of two-theta and was ~22° at 2 Å$^{-1}$ to 70° at 5 Å$^{-1}$. The XRD data were azimuthally integrated as a function of scattering angle, $2\theta$ and converted to momentum transfer, $q = 4\pi \sin \theta/\lambda$, where the wavelength for 11 keV x-rays is $\lambda = 1.127$ Å. Calibration of integrated diffraction patterns was performed using the analysis program Dioptas[67]. The Scherrer equation, $L_c = 0.92\pi/FWHM$, was used to estimate the size of crystalline domains from the width of the Bragg peaks[48].

SAXS was measured with one in-chamber CSPAD quad and two $2 \times 2$ CSPADs ($x = 90$ cm), and a single $2 \times 2$ CSPAD detector outside the chamber at $x = 2.5$ m. The coverage of the SAXS detectors was $0.5 > q > 0.02$ Å$^{-1}$ and calibrated using diffraction from an Ag behenate standard. After dark-field correction and subtraction of the associated static SAXS pattern (essentially equivalent to an empty ablator/LiF cell), the dynamic SAXS data were analyzed using an empirical Guinier–Porod fitting approach assuming dilute conditions. This model provided the fewest possible assumptions about the product morphologies and distributions. Based on Guinier's law and Porod's law, the SAXS contributions is

$$I(Q) = G \exp\left(\frac{-Q^2 R_g^2}{3}\right) \text{ for } Q \leq Q_1 \qquad (1)$$

$$I(Q) = \frac{B}{Q^P} \text{ for } Q \geq Q_1 \qquad (2)$$

where $R_g$ is the radius of gyration, $G$ is the Guinier scaling factor, $B$ is the Porod scaling factor, and $P$ is the Porod exponent. Constraining the values of these terms and their derivatives to be continuous at $Q_1$ results in the relationships:

$$Q^1 = R_g^{-1} \sqrt{\frac{3P}{2}} \qquad (3)$$

$$B = G \exp\left(-\frac{P}{2}\right)\left(\frac{3P}{2}\right)^{\frac{d}{2}} \frac{1}{R_g^P} \qquad (4)$$

thereby eliminating $P$ and $Q_1$ as fitting parameters[68]. The data for products formed

at 55 GPa could be approximated using a power-law and a constant background ($G_1 = 0$), but in most cases, the fit was marginally improved by including an additional Guinier–Porod contribution. Data for products formed at 27 GPa could be fit by either the sum of a Guinier-Porod contribution, a power-law, and a constant background or the sum of a log-normal distribution of Guinier–Porod contributions and constant background.

**Equation of state**. The EOS of unreacted (liquid) benzene was based on the Sesame framework[69], where the zero-temperature compression response was based on a fit to shock data and the thermal component on a generalized form of the Tarasov model[70]. The EOS of shock-driven decomposition products was built from thermochemical modeling using Ross perturbation theory based on exponential-6 potentials and ideal mixing. Additional details are provided in Supplementary Note 1.

## Data availability
The shock state data generated in this study are given in the main text and tables. The diffraction peaks and scattering fits are found in the main text and supplemental information. The Sesame tabular EOS are available upon request to J.D.C.

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

## Acknowledgements

This work was supported by DOE/NNSA Campaign 2 at Los Alamos National Laboratory, and was awarded under beamtime proposal "LO08: Evolution of the structure of nanocarbon in high pressure/temperature chemical reactions." We thank Cynthia A. Bolme for technical advice pertaining to the experiments. A.E.G. acknowledges support from a LANL Reines Fellowship. Use of the Linac Coherent Light Source (LCLS), SLAC National Accelerator Laboratory, is supported by the U.S. Department of Energy (DOE), Office of Science, Office of Basic Energy Sciences under Contract No. DE AC02-76SF00515. The Matter in Extreme Conditions (MEC) instrument of LCLS has additional support from the DOE, Office of Science, Office of Fusion Energy Sciences under contract No. DE-AC02-76SF00515. Los Alamos National Laboratory is operated by Triad, LLC for the U.S. Department of Energy.

## Author contributions

D.M.D., E.B.W., M.A.F., R.C.H., R.L.G., B.S.R., D.P., and R.L.S. participated in the experiments, and contributed to data analysis and interpretation. J.D.C. contributed with the equation of state of benzene. H.J.L., A.E.G., and E.G. participated in the experiment as instrument scientists at the MEC. E.B.W. served as local spokesperson for the experiment, and led the data analysis. D.M.D. was the principal investigator, and wrote the paper.

## Competing interests

The authors declare no competing interests.
