## [Peer Review File · Nature Communications]

REVIEWER COMMENTS

Reviewer #1 (Remarks to the Author):

This paper describes an ambitious experiment which obtains WAXS/SAXS data from shocked benzene, in which solid carbon is expected to condense. The authors likely observe a range of carbon polymorphs nominally identified via WAXS, and their principal conclusion is the formation of, among other polymorphs, polymerized carbon (H18) possibly as an intermediate to the ultimate formation of cubic or hexagonal diamond. Another conclusion of this work is that the chemistry of carbon under extreme conditions is very complex. Often simulations suggest a substantial number of intermediate states (even in gas phase reactions) particularly with carbon, and this work seems to confirm that.

I hesitate to recommend publication at this point because there are many questions the authors should answer to fully describe this work, and the paper presents the observation of a somewhat non-specific distribution of intermediates without providing much definitive guidance about how this observation informs our understanding of diamond formation (or some other related carbon chemistry) under shock conditions. For instance, is the observation of H18 a pivotal (perhaps necessary) step in the formation of diamond? In diamond formation in explosives? Is this the first time H18 has been observed in a dynamic experiment? Do these data suggest particular transformation mechanisms? How do these empirically suggested mechanisms influence models? Can phase fractions be obtained? Possible texturing in a liquid suggests some directional dependence in the chemistry. Can the authors elaborate on this? Why do the diffraction patterns differ so much? Are there critical phenomena in this regime?

If the authors answer specific technical questions below and better contextualize the results to indicate their specific significance to the broader science of carbon chemistry (e.g. is H18 a bottleneck in diamond formation?), this paper may be publishable in Nature Communications. I hope the authors can answer the questions below (which I believe will improve the paper) in any event.

Can the authors more definitively outline the scientific goal of this experiment? The authors may want to put the phase diagram earlier in the paper and add some justification for shocking to the thermodynamic states they obtained in the experiment (e.g. "we shocked to the diamond region of the phase diagram to observe possible intermediate states along the path to diamond formation"). Fig. 5A seems to indicate that the experiment shocked to regions of the phase diagram corresponding to carbon onions (55 GPa shots) and liquid nanocarbon (27 GPa shot), yet the authors discuss observed carbon polymorphs as intermediates to diamond formation (when it doesn't appear that diamond would be the expected final equilibrium state). Why? Also, in at least one shot,

the authors likely observe diamond. How (if they weren't shocking to the diamond region of the phase diagram)? Doesn't this cast some doubt on their assignment of the thermodynamic state? Did the authors expect to see diamond as a final product (given the phase diagram)? Why did they only see diamond in one of the 55 GPa shots?

Generally, there is a lot of variation in the WAXS results between shots. Can the authors address possible causes of this variation? All of the WAXS data is shown in the supplemental information. Why not show/discuss all of the WAXS data in the main text?

The authors mention VISAR data taken during the shot, but do not show these data. Although they may not need to show every trace, an example from their data in the supplemental information would be useful – do the authors see a steady compression or variation in the (LiF) particle speed? What rise time do they measure at the exit surface? A VISAR trace will provide quantitative answers to lots of hydro-related questions. For instance, a quasi-thermal (but not yet chemical) equilibrium may occur very quickly in benzene (ps scale, similar to other liquids). For a ns time scale pressure rise, the initial compression may be quasi-isentropic (rather than shock). What was the (pressure) rise time? The experiment may have employed a very fast laser rise, but the authors do not cite the laser rise time (as far as I can tell – it may be in cited work, but why not state it here?).

Related to this, some more description of the hydrodynamic situation would be helpful. What were the shock/particle speeds? Can these be included in Table 1? The authors mention that they took data when approximately 90% of the sample was compressed. What was the timing for this? Without seeing pressure shifts in the diffraction data, are they confident that they were observing compressed sample with the correct timing?

Did the authors see pressure/temperature shifts in the data? They cite d-spacings for the shocked sample and compare to ambient. The shocked d-spacings seem quantitatively similar to the ambient values, but since this is nominally compressed material, shouldn't the shocked d-spacings be smaller than ambient (consistent with higher density under compression)?

I didn't see any unshocked SAXS data – this might be useful for comparison to the shocked SAXS data in Fig. 2.

A Hugoniot in Fig. 5A would be very helpful. I assume the points in Fig. 5A correspond to the shocked states the authors reach in the experiment. The authors write, "The states reached in shocked benzene are within the phase stability regions where hexagonal and cubic diamond, and liquid

carbon phases might be expected to exist, Figure 5A,” but this doesn’t seem consistent with the phase diagram labels, which don’t seem to indicate diamond for the PT regions of the shots.

The authors note significant azimuthal variation in the diffraction patterns. Can they comment on possible reasons for this variation? If the sample remains liquid (does it?), I wouldn’t expect it to support anisotropic stress. Is there some preferential orientation to the products (which would be very interesting)?

Peak positions in Fig. 4 are useful, but plots of simulated diffraction patterns (including intensities) for the cited structures would be more useful. Above about 2.5 Å⁻¹, there is not much consistency between the peak intensities (and, effectively, peak positions) for different shots. The peak positions for both H18 and other carbon polymorphs may be correlated with calculated peak positions, but the peaks in the data and peaks for carbon polymorphs are densely distributed – without a comparison to peak intensities, it is difficult to assign peaks or species. This paper would benefit from plots of peak intensities and a comparison to the observed intensities which might give more confidence that the assigned identifications really do correspond to H18 and graphite, or narrow down the possibilities. Run 239 seems generally to have the best correspondence to the density of peak locations, but even in this run there is disagreement, particularly in the 4.3-4.9 Å⁻¹ range. Are the observed peaks consistent in both position and intensity? If not, why not?

Reviewer #2 (Remarks to the Author):

Review of Dattelbaum et al.

Dattelbaum et al. employ a combination of SAXS and XRD, supported by theoretical approaches to investigate the influence of high pressure shock of benzene. The manuscript is clear and well written and follows a range of other bodies of work by the team. However, I will limit my major comments to my principal area of expertise, namely small-angle scattering. In this context, I feel that the work has major shortcomings which need to be addressed.

What is the basis for assuming that the products are spherical dilute particles? Accepted that this is the simplest model that one could use but what is the reasoning behind this approach? Have the products been examined post-experiment? Is this possible? Could the fragments formed be studied with SEM (e.g. Fig. 5C)? I would have thought, given the XRD data and interpretation thereof, that

the products would more likely be elongated particles. I do wonder whether the Gaussian distribution of particles (which actually I suspect is a Schulz distribution for the fitting) is a manifestation of ellipsoidal objects of varying dimensions. Can the authors comment on this? The authors attempted to use a Guinier-Porod model (fine) but what was the intent of using a -3 Porod exponent?

As far as the modelling is concerned, I would like to have seen a table of all fitting parameters and associated errors (I will address the error notation below). I would like to see all fitting parameters and values that were constrained as well as refined. Some are tabulated here but not all. For example, was the scattering length density of the products with respect to the surrounds considered? If the data were collected on an absolute scale, one may be able to comment on particle density or SLD. As far as the fit is concerned, I will like to see it displayed more clearly in the figure. Perhaps change the size of the data points or use different symbols? To my eye, it appears that the blue scattering pattern is rather different to the other two despite the same experimental conditions yet the fit values are remarkably similar. Why is this?

The authors state: 'SAXS provides an accurate means to evaluate nanoscale product morphology': This is not really true and the word 'accurate' should be removed. It's merely a means of measurement; the morphology is 'determined' by models but the models should be backed up with complementary characterisation and this is where I feel that the work lacks depth. This is particularly the case following my familiarising myself with the associated SAS-related refs. in the manuscript where (most of) the models previously reported appear to be more robust or more justified.

I was particularly concerned about the q range over which data have been presented. The experiment was performed with multiple detectors with a reported q range of $0.02 - 0.5 \text{ \AA}^{-1}$ so where is the rest of the data?; only $0.02 - 0.07$ is shown. To show a 'fit' over such a short range fails to convince the reviewer. The authors should present all the data over the full q range and then indicate why they wish to exclude points above 0.07 if that is indeed the case. I have compared the data presented here with other related work in the manuscript and I would expect there to be information that can be retrieved at the higher scattering angles.

'Experiments delivering ca. half the input pressure (27 GPa) show nearly identical SAXS patterns (See Supplementary Information)' – no data are present in Supplementary Information. I would have liked to have seen different pressures and associated SAXS patterns to understand how they evolve under different parameters. This would lend greater credibility to the models applied and the interpretation given.

Error notation: It would be better to quote r as follows: 56(2) or 56 ± 2 ; 56(3); 56(4) etc. with the error denoted in the last significant figure. Ditto for radial spread. How does one determine the temperature as $4940 \pm 710\text{K}$? Is it better to quote 5000(700)?

Figure 3 - why is the static Au peak at 2.7 not conserved in the dynamic data? Would one not expect all Au peaks to be present and in same position?

How was L_c determined? Method / equation should be added. What is the Scherrer equation?

L_c varies from 60 to 150 Å; "crystallite sizes are consistent with the spherical particle dimensions obtained from SAXS" – how? This is not obvious. Do these values not suggest ellipsoidal or related structures?

Other comments:

- Under static compression, benzene has a limited region of liquid phase stability, and at low pressures...; should this not read 'high'?
- Figure 1 and caption – it is difficult to discern 7 detectors here. More detail is needed; I can see 6 (I think). 'place' should be 'placed'
- 'WAXS elucidates the crystal structures formed during the dynamic drive...' should be elucidates
- 'average adius' - radius
- 'Table 1 that the polydispersity suggestes' suggests
- 'shock adibat' – adiabat
- FWHM pk1.81 – better to write FWHM($q = 1.81 \text{ \AA}^{-1}$)
- weakly coupled single graphane layers in a chair conformation
- chair vs. boat – versus
- SAXS method – calibrations? dark field corrections?
- 'The SAXS data were fit using solid spheres with a Gaussaian distribution' – fitted; Gaussian

Unfortunately, given my range of concerns as to the limited SAXS q range shown, fitting method used, the lack of justification of the model, and the proposed interpretation in the context of the XRD data, I feel that the work in its current form does not merit publication. However, I suspect that the assembled team should be able to address many of these issues in a substantially revised submission.

Reviewer #3 (Remarks to the Author):

This manuscript describes formation of carbon clusters in shock-compressed benzene. The team completed very impressive investigations of chemical transformations in highly reactive environments at state of the art ultrafast XFEL facility at SLAC. The thoroughness of the analysis, the broad and meaningful interpretation set the high standard of such experiments. I strongly recommend to accept this manuscript as this is a high-impact work.

One question/suggestion: I'd like to see the dynamics of chemical transformations as reflected in evolution of the diffraction peaks. Can authors include this information and infer some stages of reactions resulting in carbon nanostructures?

Technical suggestion: quality of graphics in insert of Fig. 1 and Fig. 2 should be improved.

Explosives Science and Shock Physics Division

Los Alamos National Laboratory
Mailstop P952
Los Alamos, NM 87545
505-667-7329

Date: January 25, 2021

Manuscript “Carbon clusters formed from shocked benzene,” D. M. Dattelbaum *et al.*

Response to Reviewers’ Comments

Reviewer #1:

This paper describes an ambitious experiment which obtains WAXS/SAXS data from shocked benzene, in which solid carbon is expected to condense. The authors likely observe a range of carbon polymorphs nominally identified via WAXS, and their principal conclusion is the formation of, among other polymorphs, polymerized carbon (H18) possibly as an intermediate to the ultimate formation of cubic or hexagonal diamond. Another conclusion of this work is that the chemistry of carbon under extreme conditions is very complex. Often simulations suggest a substantial number of intermediate states (even in gas phase reactions) particularly with carbon, and this work seems to confirm that.

I hesitate to recommend publication at this point because there are many questions the authors should answer to fully describe this work, and the paper presents the observation of a somewhat non-specific distribution of intermediates without providing much definitive guidance about how this observation informs our understanding of diamond formation (or some other related carbon chemistry) under shock conditions. For instance, is the observation of H18 a pivotal (perhaps necessary) step in the formation of diamond?

To address comment 1, we have added text to the introduction to better describe the aims of the experiment and principal conclusions. The observation of H18 is a key step in the formation of diamond, and the first observation in shocked benzene. The sp²-sp³ layered structure appears to be an intermediate in formation of diamond.

In diamond formation in explosives?

We cannot comment here on diamond formation in explosives as the experiments were on benzene. From our time-resolved SAXS experiments on explosives, we do see layered structures and onion-like particle formation, and we have referenced those papers in this paper.

Is this the first time H18 has been observed in a dynamic experiment? *Yes, to our knowledge. However, graphite and diamond have been observed by Kraus et al. And sp²-sp³ carbon forms have been observed in recovered detonation products (this group, and others) and in pressure-shear diamond cells (Blank et al.)*

Do these data suggest particular transformation mechanisms? How do these empirically suggested mechanisms influence models? *See Figure 5. It suggests that benzene rings condense into a layered sheet, that is then de-hydrogenated and further condensed into diamond. This mechanism and regions of*

(meta)stability in the carbon phase diagram will support kinetics models of chemical transformations in shocks, including reactive burn models for explosives.

Can phase fractions be obtained? *Because the experiments were powder diffraction, the detectors did not cover the full azimuthal range of the diffraction, and because the experiment is a uniaxial compression geometry, phase fractions cannot be obtained from the recorded patterns.*

Possible texturing in a liquid suggests some directional dependence in the chemistry. (some texture in pdts) Can the authors elaborate on this? Why do the diffraction patterns differ so much? Are there critical phenomena in this regime?

There is no observed texturing in the liquid prior to the shock from the diffuse scattering observed in the pre-shot static data. The compression experiment is uniaxial and there does appear to be non-uniform texturing observed in the products, though we believe this is due to the layered structures forming as benzene molecules “zip” together to form the layers within the carbon particles. The patterns differ mostly in the recorded intensities, not in the d-spacings of the reflections. We believe this is due to catching metastable states along the evolution of product formation.

If the authors answer specific technical questions below and better contextualize the results to indicate their specific significance to the broader science of carbon chemistry (e.g. is H18 a bottleneck in diamond formation?), this paper may be publishable in Nature Communications. I hope the authors can answer the questions below (which I believe will improve the paper) in any event.

Can the authors more definitively outline the scientific goal of this experiment? *We have added more details to the introduction on the goal of the experiment.*

The authors may want to put the phase diagram earlier in the paper and add some justification for shocking to the thermodynamic states they obtained in the experiment (e.g. “we shocked to the diamond region of the phase diagram to observe possible intermediate states along the path to diamond formation”). *We have edited Figure 5 to include a better representation of the carbon phase diagram from our own and literature work.*

Fig. 5A seems to indicate that the experiment shocked to regions of the phase diagram corresponding to carbon onions (55 GPa shots) and liquid nanocarbon (27 GPa shot), yet the authors discuss observed carbon polymorphs as intermediates to diamond formation (when it doesn't appear that diamond would be the expected final equilibrium state). *We have added the equilibrium phase boundaries to Figure 5. Why? Also, in at least one shot, the authors likely observe diamond. How (if they weren't shocking to the diamond region of the phase diagram)? Doesn't this cast some doubt on their assignment of the thermodynamic state? Did the authors expect to see diamond as a final product (given the phase diagram)? Why did they only see diamond in one of the 55 GPa shots?*

The carbon phase diagram included in the original manuscript has been replaced by the more traditional one, corroborated by many more sources over many decades. Diamond is the expected equilibrium state in both shots. Failure to observe diamond in all of the 55 GPa shots is most likely just a reflection of kinetic limitations. Some of the possible sources of variation between shots at nominally the same pressure are discussed in the next point.

Generally, there is a lot of variation in the WAXS results between shots. Can the authors address possible causes of this variation? *The variations in the WAXS results are likely due to probing evolving, metastable states, and variation in the average P/T conditions that the XFEL is probing (sample volume, local distribution of products, and products evolving for different time periods).* All of the WAXS data is shown in the supplemental information. Why not show/discuss all of the WAXS data in the main text? *We chose 3 representative patterns as the others were similar, and included them in the supplemental. All data is presented.*

The authors mention VISAR data taken during the shot, but do not show these data. Although they may not need to show every trace, an example from their data in the supplemental information would be useful – do the authors see a steady compression or variation in the (LiF) particle speed? *Yes, the shock is sustained for 20 ns.*

What rise time do they measure at the exit surface? *~ 1 ns risetime measured.*

A VISAR trace will provide quantitative answers to lots of hydro-related questions. For instance, a quasi-thermal (but not yet chemical) equilibrium may occur very quickly in benzene (ps scale, similar to other liquids). For a ns time scale pressure rise, the initial compression may be quasi-isentropic (rather than shock). What was the (pressure) rise time? *(<1 ns based on the measured shock wave profiles)*

The experiment may have employed a very fast laser rise, but the authors do not cite the laser rise time (as far as I can tell – it may be in cited work, but why not state it here?). *We have chosen to include a representative VISAR profile to the Supplementary material to address these questions.*

Related to this, some more description of the hydrodynamic situation would be helpful. What were the shock/particle speeds? *These have been added as a footnote to the Table.*

Can these be included in Table 1? The authors mention that they took data when approximately 90% of the sample was compressed. What was the timing for this? *We did not want to convolute double shock states, so we left some unshocked material from known timing through sample.*

Without seeing pressure shifts in the diffraction data are they confident that they were observing compressed sample with the correct timing? *The initial sample is liquid and the experiments were timed so that the shockwave travels through ~90% of the benzene before reaching the back window. The timing was confirmed with over 200 experiments in the same run including those on ablator/window (no sample) configurations, and with the measured VISAR signals showing the wave reaching the back window relative to the XFEL pulse.*

Did the authors see pressure/temperature shifts in the data? *We did not observe this due to the averaging (through sample) and single event nature of the measurements.*

They cite d-spacings for the shocked sample and compare to ambient. The shocked d-spacings seem quantitatively similar to the ambient values, but since this is nominally compressed material, shouldn't the shocked d-spacings be smaller than ambient (consistent with higher density under compression)? *Pressure compresses the lattice but temperature can expand the lattice, so it isn't clear what the best reference unit cell for a given phase is, and the proposed phases have not all been observed under shock loading which differs*

from static compression. In the previous version, the Graphate lines in the figure and in the SI tables were from a 50 GPa and ambient T calculation (provided by Wen). The H18 lines are shown compressed to 50 GPa (still ambient T) according to the 360 GPa bulk modulus. Also cubic diamond lines at ambient P and 50 GPa (both ambient T) are shown.

I didn't see any unshocked SAXS data – this might be useful for comparison to the shocked SAXS data in Fig. 2.

We agree with the reviewer that this is a useful comparison for the reader and have included the static (unshocked) SAXS data in the figure. While the dynamic SAXS data for run 303 and 237 are shifted vertically (for clarity), the static data (grey) is on the same intensity scale as dynamic run 239 (red) to enable a 1-to-1 comparison that shows the static SAXS intensity is an order of magnitude weaker than the SAXS signal from the reaction products.

A Hugoniot in Fig. 5A would be very helpful. I assume the points in Fig. 5A correspond to the shocked states the authors reach in the experiment. The authors write, “The states reached in shocked benzene are within the phase stability regions where hexagonal and cubic diamond, and liquid carbon phases might be expected to exist, Figure 5A,” but this doesn't seem consistent with the phase diagram labels, which don't seem to indicate diamond for the PT regions of the shots.

Unreacted and products Hugoniots were added to Figure 5. The states reached in the experiments are shown with their error bars as described in the Supplemental in Figure 5. We have re-done the carbon phase diagram and it is consistent with several literature references, shown and annotated in the Figure.

The authors note significant azimuthal variation in the diffraction patterns. Can they comment on possible reasons for this variation? If the sample remains liquid (does it?), I wouldn't expect it to support anisotropic stress. Is there some preferential orientation to the products (which would be very interesting)?

Peak positions in Fig. 4 are useful, but plots of simulated diffraction patterns (including intensities) for the cited structures would be more useful.

We have revised Fig. 4 for clarity, but have not provided the calculated patterns. The calculated patterns shown in the Figure below are broadened with a 0.05 \AA^{-1} FWHM and are consistent with the data. All calculated patterns are at 50 GPa. Cubic diamond lines are ambient (lower q ones) and at 50 GPa (higher q ones). We have chosen not to include the calculated patterns due to the differences between the shock environment with metastable products and calculated static patterns.

Above about 2.5 \AA^{-1} , there is not much consistency between the peak intensities (and, effectively, peak positions) for different shots. The peak positions for both H18 and other carbon polymorphs may be correlated with calculated peak positions, but the peaks in the data and peaks for carbon polymorphs are densely distributed – without a comparison to peak intensities, it is difficult to assign peaks or species. This paper would benefit from plots of peak intensities and a comparison to the observed intensities which might give more confidence that the assigned identifications really do correspond to H18 and graphate, or narrow down the possibilities. Run 239 seems generally seems to have the best correspondence to the density of peak locations, but even in this run there is disagreement, particularly in the 4.3-4.9 \AA^{-1} range. Are the observed peaks consistent in both position and intensity? If not, why not?

In short, the peak intensities are not entirely consistent for the reasons described above. We chose a CH and C phase that were representative of the peak positions. At best, we are showing that peaks that can't be indexed to a conventional C phase (i.e. graphite, hex D, cubic D) do not reproduce the scattering pattern. Peak intensities are an entirely different problem: we do not measure the full azimuthal range of the diffraction and the uniaxial compression breaks the symmetry of the problem (i.e. there is texture) so we can't be quantitative when it comes to relative peak intensities.

Reviewer #2 (Remarks to the Author):

Dattelbaum et al. employ a combination of SAXS and XRD, supported by theoretical approaches to investigate the influence of high pressure shock of benzene. The manuscript is clear and well written and follows a range of other bodies of work by the team. However, I will limit my major comments to my principal area of expertise, namely small-angle scattering. In this context, I feel that the work has major shortcomings which need to be addressed.

What is the basis for assuming that the products are spherical dilute particles? Accepted that this is the simplest model that one could use but what is the reasoning behind this approach?

We want to stress that the single-event nature of the experiment and the particular challenges of performing SAXS measurements on the MEC beamline limit the richness of information that we are able to extract. Our analysis using spherical dilute particles was intended as a simplifying assumption, as the reviewer surmises, and we did not intend to give the impression that the SAXS data conclusively identified a spherical morphology of the products. In the revision, we have made significant efforts to readdress the SAXS analysis (further details given in the following responses). This has included changing from a spherical form factor model to interpret the data using an empirical Guinier-Porod model that does not make any assumptions about particle shape. Using this model, we extract length scales associated with the reaction products (in terms of the radius of gyration, R_g) but do not impose any assumption about the product morphology.

Have the products been examined post-experiment?

No, in the experiment, multiple samples are loaded onto the sample holder in the vacuum chamber at MEC. The laser ablation/shock process destroys the targets, making recovery impossible in the experimental configuration which is designed for in situ measurements. Note that we discuss historical recovery experiments by Dremine et al. However, in recovery experiments, the final products have seen complex loading/unloading pathways and temperatures.

Is this possible? Could the fragments formed be studied with SEM (e.g. Fig. 5C)?

It has been done so in the past for some materials, but again, the sample history is not known due to how recovery is performed.

I would have thought, given the XRD data and interpretation thereof, that the products would more likely be elongated particles. I do wonder whether the Gaussian distribution of particles (which actually I suspect is a Schulz distribution for the fitting) is a manifestation of ellipsoidal objects of varying dimensions. Can the authors comment on this? The authors attempted to use a Guinier-Porod model (fine) but what was the intent of using a -3 Porod exponent?

We interpret this comment of the reviewer to suggest that the layered structure observed in XRD would break the spherical symmetry of the products and result in elongated particles. This is a possible outcome, although spherical carbon particles with an internal layered structure have been observed previously. We agree that an elongated particle morphology could be modelled by spherical particles with a distribution of radii through the inclusion of additional fitting parameters (i.e. higher complexity of the model), although assuming the distribution is not bimodal this would suggest a rather low aspect ratio. However, we believe that with our revised SAXS analysis these concerns no longer apply.

As far as the modelling is concerned, I would like to have seen a table of all fitting parameters and associated errors (I will address the error notation below). I would like to see all fitting parameters and values that were constrained as well as refined. Some are tabulated here but not all. For example, was the scattering length density of the products with respect to the surrounds considered? If the data were collected on an absolute scale, one may be able to comment on particle density or SLD.

We have expanded the table to include all relevant fitting parameters (including fixed parameters) with the exception of the constant background and the scale factor associated with the low-q power law. We did not find the power-law scale factor to be relevant since it was not possible to reliably put the data on an absolute scale. Additionally, we have quantified the parameter errors using a $\chi^2 < 1.1 \chi^2_{\min}$ and

included the symmetric or asymmetric errors in the table. A table including the SAXS fitting parameters for the alternative log-normal distribution has been included in the SI.

As far as the fit is concerned, I will like to see it displayed more clearly in the figure. Perhaps change the size of the data points or use different symbols? To my eye, it appears that the blue scattering pattern is rather different to the other two despite the same experimental conditions yet the fit values are remarkably similar. Why is this?

We have improved the visual clarity of the figure to make the model intensity clear and have added the individual components of the fit as separate lines. In our revised SAXS data reduction and analysis (see responses below), we did indeed see small differences between the scattering patterns, including in run 237 (blue). These differences could be captured by small differences in the low-q power law dependence of the data.

The authors state: ‘SAXS provides an accurate means to evaluate nanoscale product morphology’: This is not really true and the word ‘accurate’ should be removed. It’s merely a means of measurement; the morphology is ‘determined’ by models but the models should be backed up with complementary characterisation and this is where I feel that the work lacks depth. This is particularly the case following my familiarising myself with the associated SAS-related refs. in the manuscript where (most of) the models previously reported appear to be more robust or more justified.

We agree with the reviewer’s comment and have removed the word ‘accurate’.

Indeed, complementary characterization can be very valuable to guide and support the structural interpretation of the products obtained by SAXS modelling. However, it was not possible to obtain such complementary data in the in-situ dynamic experiments described here and the transient nature of the structures preclude direct comparison with recovered materials. In the absence of such complementary information, our approach was to apply the simplest possible model that could reproduce the features of the SAXS data and yield a low χ^2 goodness-of-fit value.

I was particularly concerned about the q range over which data have been presented. The experiment was performed with multiple detectors with a reported q range of 0.02 – 0.5 Å⁻¹ so where is the rest of the data?; only 0.02 – 0.07 is shown. To show a ‘fit’ over such a short range fails to convince the reviewer. The authors should present all the data over the full q range and then indicate why they wish to exclude points above 0.07 if that is indeed the case. I have compared the data presented here with other related work in the manuscript and I would expect there to be information that can be retrieved at the higher scattering angles.

Initially, we found that spatial intensity variations in one of the higher q SAXS detectors (CSPAD quad) resulted in significant artifacts in the SAXS intensity that prevented merging the low ($q < 0.07 \text{ \AA}^{-1}$) and high ($q > 0.05 \text{ \AA}^{-1}$) regions of the measurements. In our efforts to address this comment we reexamined all of the SAXS data and the data reduction approach and found that the smaller 2x2 CSPAD detectors did not suffer from the same intensity variations. This enabled us to extend the range of the SAXS measurement presented to the full accessible q-range, reaching instrumental background at $q \sim 0.4 \text{ \AA}^{-1}$. During this process, we also identified and remedied a systematic error in the low-q region of the SAXS measurement. As a result, all

the SAXS data has been reanalyzed to model the scattering over an approximately order of magnitude greater q -range than we had previously achieved.

‘Experiments delivering ca. half the input pressure (27 GPa) show nearly identical SAXS patterns (See Supplementary Information)’ – no data are present in Supplementary Information. I would have liked to have seen different pressures and associated SAXS patterns to understand how they evolve under different parameters. This would lend greater credibility to the models applied and the interpretation given.

Per the reviewer’s suggestion, the SAXS data, analysis, and description of the 27 GPa measurement have been added to the main text. We find a significant difference in the SAXS data at lower input pressure associated with an approximately 4 nm length scale associated with the reaction products. Such a length scale was not identified within the products formed at 55 GPa, although there was some evidence for a smaller, ~1 nm, length scale associated with the products formed under these higher input pressure conditions. It also suggests that at 55 GPa there exists a length scale associated with the products that is larger than the accessible SAXS measurement window, greater than ~8nm.

Error notation: It would be better to quote r as follows: 56(2) or 56 \pm 2; 56(3); 56(4) etc. with the error denoted in the last significant figure. Ditto for radial spread. *Appropriate error notation has been included in the table.*

How does one determine the temperature as 4940 \pm 710K ? Is it better to quote 5000(700) ?
Point taken on the accuracy; however we gave the calculated temperatures of the average between the reactants and products Hugoniot, using high fidelity equations of state as described in the Supplementary material. We further added those plots to Figure 1.

Figure 3 - why is the static Au peak at 2.7 not conserved in the dynamic data? Would one not expect all Au peaks to be present and in same position?
The thin gold coating was vaporized during the ablation/shock generation in the experiments. It was added in the hopes we could obtain in situ pressure or temperature measurements but it did not work out.

How was L_c determined? Method / equation should be added. What is the Scherrer equation?
Yes, the Scherrer equation was used to estimate L_c . The Scherrer equation reference was added to the text and is now given in the Methods section.

L_c varies from 60 to 150 Å; “crystallite sizes are consistent with the spherical particle dimensions obtained from SAXS” – how? This is not obvious. Do these values not suggest ellipsoidal or related structures?

We have revised the text about this point, with the new SAXS analysis, we can’t make this statement. In general, the L_c based on the Bragg peak FWHM is larger than the SAXS window.

Other comments:

The following typographical errors have been addressed.

- Under static compression, benzene has a limited region of liquid phase stability, and at low pressures...;

should this not read 'high'?

- Figure 1 and caption -More detail is needed; I can see 6 (I think). 'place' should be 'placed'
- 'WAXS elucidates the crystal structures formed during the dynamic drive...' should be elucidates
- 'average adius' - radius
- 'Table 1 that the polydispersity suggestes' suggests
- 'shock adibat' – adiabat
- FWHM pk1.81 – better to write FWHM ($q = 1.81 \text{ \AA}^{-1}$)
- weakly coupled single graphane layers in a chair conformation
- chair vs. boat – versus
- SAXS method – calibrations ? dark field corrections ?

Details on the dark-field correction, subtraction of scattering from the empty cell, and methods for detector calibration have been added to the methods.

- 'The SAXS data were fit using solid spheres with a Gaussaian distribution' – fitted; Gaussian

Unfortunately, given my range of concerns as to the limited SAXS q range shown, fitting method used, the lack of justification of the model, and the proposed interpretation in the context of the XRD data, I feel that the work in its current form does not merit publication. However, I suspect that the assembled team should be able to address many of these issues in a substantially revised submission.

We thank the reviewer for the detailed criticism of the SAXS component of the manuscript. We believe that we have thoroughly addressed his/her main points by (1) emphasizing the experimental challenges involved in performing SAXS measurements of single-events, (2) extending the SAXS q range by approximately an order of magnitude, and (3) implementing a new empirical fitting method with fewer a priori assumptions than the previous spherical form factor-based interpretation. We believe these changes have strongly improved the manuscript and are grateful to the reviewer for the suggestions.

Reviewer #3 (Remarks to the Author):

This manuscript describes formation of carbon clusters in shock-compressed benzene. The team completed very impressive investigations of chemical transformations in highly reactive environments at state-of-the-art ultrafast XFEL facility at SLAC. The thoroughness of the analysis, the broad and meaningful interpretation set the high standard of such experiments. I strongly recommend to accept this manuscript as this is a high-impact work.

One question/suggestion: I'd like to see the dynamics of chemical transformations as reflected in evolution of the diffraction peaks. Can authors include this information and infer some stages of reactions resulting in carbon nanostructures?

This is a very interesting idea. We went back and looked at the timing of the XFEL measurement vs. shock breakout at the benzene/LiF window. There is a trend in the XRD data with run 303 looking like a mixture of run 239 and 294. There are also signs of a trend there in the SAXS data, the power-law increases with increasing time. We can imagine this as the Porod component of a Guinier region outside our measurement window (bigger than $R_g=8\text{nm}$) that is evolving towards smooth surfaced particles. For now, due to the single event nature of the experiment, we have decided not to include this in the text.

Technical suggestion: quality of graphics in insert of Fig. 1 and Fig. 2 should be improved.

The resolution of the images will be improved once the manuscript is accepted.

Reviewer #1 (Remarks to the Author):

This revision is a substantial improvement to the original, including many more experimental and analysis details. I don't feel this paper strongly supports a particular transformation mechanism to diamond, but does support the existence of unusual carbon species which are candidates as intermediates to diamond. Due to the complexity of the diffraction patterns (and their shot to shot variation), it may not be appropriate to strongly assert the observation of any dominant intermediate, yet the data unambiguously show solid condensates which are disordered carbon. At the very least, these results establish a lower bound for the time scale of formation of equilibrium products under the compression conditions.

I do not oppose publication of this paper, but I still can't support it before the authors address issues below.

These results purport to support the incorporation of hydrogen in solid products at the observation time, although congestion in the diffraction patterns and uncertainty about pressure/temperature shifts make this case more difficult. Although the authors cite possible matches in line positions to hydrogen-containing species, they do not present analysis with other candidate species, such as the many forms of moderately disordered pure carbon which might also be present, and can exhibit many more lines than (more symmetric) graphite or diamond. It is worth noting that simulated patterns of Graphate I,II and H18 at the bottom of Fig. 4A do not (aside from interlayer reflections just below 2.0 \AA^{-1}) show strong correlation with the experimental patterns. Surprisingly, the authors do not compare line positions with respect to ordinary graphite. Many lines which the authors assign to Graphate or H18 might also be explained with pressure-shifted graphite lines (to within errors the authors seem to accept based on other assignments). For instance:

Run 303 Observed (\AA^{-1})	Graphite ijk	Graphite (\AA^{-1})
1.81	002	1.876698
1.87	002	1.876698
2.44	N/A	N/A
2.99	100	2.954068
3.58	102?	3.499797
3.74	N/A	N/A
3.89	N/A	N/A
4.12	103	4.080573
4.43	N/A	N/A
5.04	110	5.1166

So, although hydrogen containing products might be expected and have experimental support, if the authors did not include pure carbon lines in their analysis, it's not clear that forms of pure carbon wouldn't correlate with many of the experimental lines. Granted, there are lines that may be unique to hydrogen-containing phases, but if the authors did not even consider graphite as a possible product, there may be less symmetric forms of pure carbon which might explain the observed anomalous lines, particularly since these are intermediate forms.

This is exacerbated by a lack of clarity in the text over whether the authors used pressure-shifted lines for comparison – see other comments below. These pressure shifts are critical for comparisons of phases with complex structures, since patterns with many lines may see correlation of a given line with different species depending on the pressure (or even not depending on temperature, as the author’s tables illustrate).

It’s unreasonable to ask the authors to consider every possible form of carbon, but excluding the identification of graphite from the analysis and then later claiming that graphite is not a major constituent is not consistent. I’d be happy if the authors included graphite as a possible candidate in their comparison tables and some caveat statement that not all (of many) carbon forms could be compared to the data. It would help the paper if the authors also identified lines (in Fig. 4, perhaps?) which seem exclusively identified with hydrogen containing forms.

I find the author’s assertion: “The solid carbon product composition is complex, and indicates that neither diamond or graphite is the major constituent, illustrating the limitation of thermochemical equilibrium approaches to modeling of decomposition products” to be inconsistent with the data. Many lines (see above) seem to overlap graphite, so it’s not clear to me how the authors can assert that graphite is not a major constituent without at least some compositional analysis. Identification of lines is not enough. Further, at the very least, the authors must qualify this statement given the time scale of the observation and transformation kinetics. This experiment does not test the distribution of products given by thermochemical calculations at equilibrium since equilibrium is never reached in the experiment. A better way to say it would be: “thermochemical equilibrium approaches to modeling of decomposition products do not work at time scales less than such-and-so (from this study) due to slow carbon kinetics” – still a significant result of this work.

The authors provide useful explanation of the SAXS data, but the conclusions are not clearly summarized. The SAXS section would benefit from a straightforward statement along the lines of: “The data at [pressure] [expression of confidence, e.g. “strongly indicate” or “mildly suggest”] the formation of [size] nm products.” My interpretation of the current revision is that the authors are confident they observed 4-5 nm products in the 27 GPa data, but less confident about the particle size observation at 55 GPa, suggesting that the product size may be outside the observation window (i.e. >50 nm in the main text or >8 nm in the response letter), or, from fits, 1 nm. This is a large size range, so the authors may want to provide some more summary guidance as to which is more likely, or otherwise express some confidence (or lack thereof) in the results. If this could be quantified with statistics, that would be preferred. The authors also cite >8 nm as the window limit in their response, so it’s not clear which is the actual window limit.

The authors may want to add comments about randomly induced texture of products (and limited azimuthal observation range) as an explanation for why the integrated shot-to-shot diffraction patterns vary.

I found the author’s response to my question “Did the authors see pressure/temperature shifts in the data?” to be unsatisfactory:

Author response: We did not observe this due to the averaging (through sample) and single event nature of the measurements.

It is not clear to me how the single shot nature of the experiment or averaging would eliminate pressure and temperature shifts from the data (which are universally observed in comparable experiments elsewhere), and seems particularly inconsistent with steady thermodynamic conditions (which seem to be implied by VISAR data, although it is not clear whether the VISAR data shown in Figure S2 extend to the observation time). This response also seems inconsistent with the author's later assertion in the response letter that they did use pressure shifted line positions (albeit calculated or measured at ambient temperature) for comparison to observed line positions.

The conditions corresponding to the line positions that were used for comparison to the observed lines should be stated explicitly in the captions of the tables which show the comparisons, e.g. "Graphate lines were taken from Wen et al., calculated at 50 GPa and ambient temperature." Similar for calculated patterns at the bottom of Fig. 4A.

Yakusheva et al. is cited as ref. 24, but as far as I can tell ref. 24 is not Yakusheva et al., nor is this reference in the bibliography.

Reviewer #2 (Remarks to the Author):

I appreciate the efforts of Dattelbaum et al in addressing the concerns raised in the review of their submission. While many comments have been addressed, I remain concerned about the SAXS analysis.

While the q range has been expanded, this should place even greater rigour on the modelling of the data. The approach has changed from a straightforward spherical model (where fit parameters are directed related to structure e.g. radius of gyration, contrast etc.) to a Guinier-Porod model and background with, in some cases, a two-layer Guinier-Porod model.

Unquestionably, the models appear to fit the data but the table of fitting parameters and uncertainties remain of concern and do not add much structural information. In fact, the Guinier-Porod model has been described as the model to use when one does not have an appropriate model.

The criticism of such an approach is that the extracted parameters are often not used to elucidate anything structural and are usually merely quoted as fitting parameters, as they are here. In principle, detailed information can be obtained from such a model by drilling down further but, given the uncertainty in them here, this would not be possible. For example, Table 1, the fitting uncertainty on G_2 is extremely large in all cases except for #237 which leads one to conclude that the fits are not particularly robust. Ditto Table S4 for G_1 . In my view, it is simply large objects that are generated whose size is outside the measurement window.

Other comment: Where is the fitted data for #294 in Figure 2? The parameters are listed for the fit but are not shown.

In addition, the authors state that they have employed time-resolved SAXS but there is no time evolution of data and associated fits illustrated as a function of time.

In summary, as far as the SAXS component of the manuscript is concerned, while the experimenters should be congratulated for conducting such a complex experiment, I have strong doubts about the utility of the technique as applied here.

RESPONSE TO REVIEWER 1

This revision is a substantial improvement to the original, including many more experimental and analysis details. I don't feel this paper strongly supports a particular transformation mechanism to diamond, but does support the existence of unusual carbon species which are candidates as intermediates to diamond. Due to the complexity of the diffraction patterns (and their shot to shot variation), it may not be appropriate to strongly assert the observation of any dominant intermediate, yet the data unambiguously show solid condensates which are disordered carbon. At the very least, these results establish a lower bound for the time scale of formation of equilibrium products under the compression conditions. I do not oppose publication of this paper, but I still can't support it before the authors address issues below.

We thank the reviewer for these comments and fully agree with their assessment of the results, these are indeed the salient points that we are trying to communicate in this manuscript. We have made additional efforts to communicate the key findings while emphasizing the limitations of these measurements (e.g. inability to identify a specific transformation mechanism or intermediate structure). In particular, we have clarified that the disordered carbon or hydrocarbon intermediates described in the manuscript (e.g. H18, graphane) are representative structures intended to illustrate the complexity of the solid product mixture.

These results purport to support the incorporation of hydrogen in solid products at the observation time, although congestion in the diffraction patterns and uncertainty about pressure/temperature shifts make this case more difficult. Although the authors cite possible matches in line positions to hydrogen-containing species, they do not present analysis with other candidate species, such as the many forms of moderately disordered pure carbon which might also be present, and can exhibit many more lines than (more symmetric) graphite or diamond.

We have significantly expanded the comparison of the low q diffraction peak to those of graphite and disordered graphite-like structures, explicitly taking into account predicted P-T shifts of the diffraction peaks, to better present the case for excluding graphite-like structures in the products. The moderately disordered carbon analogs to graphite and diamond have qualitatively similar diffraction patterns. We have clarified in the text the regions in q space where the measured peaks can not be indexed by such structures.

It is worth noting that simulated patterns of Graphate I,II and H18 at the bottom of Fig. 4A do not (aside from interlayer reflections just below 2.0 \AA^{-1}) show strong correlation with the experimental patterns.

We agree with the reviewer that we were unable to identify a candidate structure that had strong correlation with the measured data. It is our interpretation that this is due to the complex mixture of structures within the products. The utility of comparisons to graphate and H18 come from peaks that correlate to the measured data in regions where there are no peaks from more conventional graphite or diamond structures.

Surprisingly, the authors do not compare line positions with respect to ordinary graphite. Many lines which the authors assign to Graphate or H18 might also be explained with pressure-shifted graphite lines (to within errors the authors seem to accept based on other assignments).

Per the reviewer's suggestion, we have included the P and T shifted graphite lines in the XRD figure (4B). Indeed, a subset of the measured peaks at higher q values can be attributed to a graphitic structure. We have added to the text that these peaks can not be unambiguously indexed. In the supplementary tables, we have limited our comparison to the graphate and H18 forms because they are capable of reproducing the low- q XRD peak and to cubic and hexagonal diamond forms which may contribute to the XRD pattern without a peak at low q . This has been clarified in the text.

For instance:

Run 303 Observed (A-1) Graphite ijk Graphite (A-1)

1.81	002	1.876698
1.87	002	1.876698
2.44	N/A	N/A
2.99	100	2.954068
3.58	102?	3.499797
3.74	N/A	N/A
3.89	N/A	N/A
4.12	103	4.080573
4.43	N/A	N/A

So, although hydrogen containing products might be expected and have experimental support, if the authors did not include pure carbon lines in their analysis, it's not clear that forms of pure carbon wouldn't correlate with many of the experimental lines. Granted, there are lines that may be unique to hydrogen-containing phases, but if the authors did not even consider graphite as a possible product, there may be less symmetric forms of pure carbon which might explain the observed anomalous lines, particularly since these are intermediate forms.

We hope that the improved description of our analysis of the low- q peak better communicates how graphite was excluded as a possible product. We have also clarified that a subset of the higher- q peaks are consistent with graphite but that the expanded d-spacing indicated by the low- q peak rules out this possibility.

This is exacerbated by a lack of clarity in the text over whether the authors used pressure-shifted lines for comparison – see other comments below. These pressure shifts are critical for comparisons of phases with complex structures, since patterns with many lines may see correlation of a given line with different species depending on the pressure (or even not depending on temperature, as the author's tables illustrate).

We have clarified the P-T conditions used for all of the calculated XRD patterns used for comparison in the manuscript.

It's unreasonable to ask the authors to consider every possible form of carbon, but excluding the identification of graphite from the analysis and then later claiming that graphite is not a major constituent is not consistent.

We would like to clarify that graphite was not excluded for the analysis of the diffraction data. Indeed, there are several higher q peaks (e.g. $q \sim 3 \text{ \AA}^{-1}$) that are ambiguous and could potentially be assigned to graphite. However, we were able to eliminate graphite as a likely major constituent of the products by the lack of a low- q peak consistent with the expected P-T shift of graphite's 002 reflection.

I'd be happy if the authors included graphite as a possible candidate in their comparison tables and some caveat statement that not all (of many) carbon forms could be compared to the data. It would help the paper if the authors also identified lines (in Fig. 4, perhaps?) which seem exclusively identified with hydrogen containing forms.

The caveat and rationale behind the carbon/hydrocarbon forms we used for qualitative comparison to the data has been added. We have made additions to call out in the text the regions in Q where measured peaks were exclusively identified with either hydrogenated graphite or mixed hybridization carbon forms.

I find the author's assertion: "The solid carbon product composition is complex, and indicates that neither diamond or graphite is the major constituent, illustrating the limitation of thermochemical equilibrium approaches to modeling of decomposition products" to be inconsistent with the data. Many lines (see above) seem to overlap graphite, so it's not clear to me how the authors can assert that graphite is not a major constituent without at least some compositional analysis. Identification of lines is not enough. Further, at the very least, the authors must qualify this statement given the time scale of the observation and transformation kinetics. This experiment does not test the distribution of products given by thermochemical calculations at equilibrium since equilibrium is never reached in the experiment.

We thank the reviewer for this comment, indeed these results do not determine the equilibrium product structure. We have added text to qualify these conclusions relevant to the time scale of the observation.

A better way to say it would be: "thermochemical equilibrium approaches to modeling of decomposition products do not work at time scales less than such-and-so (from this study) due to slow carbon kinetics" – still a significant result of this work.

The authors provide useful explanation of the SAXS data, but the conclusions are not clearly summarized. The SAXS section would benefit from a straightforward statement along the lines of: "The data at [pressure] [expression of confidence, e.g. "strongly indicate" or "mildly suggest"] the formation of [size] nm products." My interpretation of the current revision is that the authors are confident they observed 4-5 nm products in the 27 GPa data, but less confident about the particle size observation at 55 GPa, suggesting that the product size may be outside the observation window (i.e. >50 nm in the main text or >8 nm in the response letter), or, from fits, 1 nm. This is a large size range, so the authors may want to provide some more summary guidance

as to which is more likely, or otherwise express some confidence (or lack thereof) in the results. If this could be quantified with statistics, that would be preferred.

We have clarified the results of the SAXS analysis, emphasizing that at 55GPa the dominant length scale corresponds to $R_g > 15\text{nm}$ (outside the measurement window) and that at 27 GPa a 5nm length scale is observed.

The authors also cite $>8\text{ nm}$ as the window limit in their response, so it's not clear which is the actual window limit.

We had made an error in the upper limit of the measurement window in the response. In the text we had used a value of 50 nm based on a simple $2\pi/q_{\min}$ approximation. In the current version, we have been more quantitative by determining the maximum R_g value that resulted in a significant deviation from a low- q power law dependence to the scattering using the Guinier-Porod model we have employed to analyze the data. We have revised our definition of the measurement's upper limit to be $R_g = 15\text{nm}$.

The authors may want to add comments about randomly induced texture of products (and limited azimuthal observation range) as an explanation for why the integrated shot-to-shot diffraction patterns vary.

I found the author's response to my question "Did the authors see pressure/temperature shifts in the data?" to be unsatisfactory:

Author response: We did not observe this due to the averaging (through sample) and single event nature of the measurements.

It is not clear to me how the single shot nature of the experiment or averaging would eliminate pressure and temperature shifts from the data (which are universally observed in comparable experiments elsewhere), and seems particularly inconsistent with steady thermodynamic conditions (which seem to be implied by VISAR data, although it is not clear whether the VISAR data shown in Figure S2 extend to the observation time).

We apologize for the misinterpretation of the reviewer's previous question. We mistakenly interpreted the question as asking if we observed a series of peak positions due to changing P/T conditions *within* a single measurement (or series of measurements). Indeed the diffraction peaks observed were shifted relative to ambient conditions due to the high pressure/temperature conditions of the measured state. We have made significant efforts to clarify this in the manuscript, particularly how the impact of P/T shifts in the peak position help to rule out graphite as a significant component of the products.

Our previous response was aimed to address the fact the shock takes several ns to propagate through the sample which results in a range of P/T states (with likely significant T variation depending on the duration of time that a given region of the sample has been shocked). Since the single-event x-ray measurement averages all of these states, we are unable to observe P/T shifts in the diffraction peaks *within* a single measurement.

This response also seems inconsistent with the author's later assertion in the response letter that they did use pressure shifted line positions (albeit calculated or measured at ambient temperature) for comparison to observed line positions.

We thank the reviewer for these comments and believe that our clarifications on the P-T shifts of the calculated patterns for comparison have greatly improved the manuscript.

The conditions corresponding to the line positions that were used for comparison to the observed lines should be stated explicitly in the captions of the tables which show the comparisons, e.g. "Graphate lines were taken from Wen et al., calculated at 50 GPa and ambient temperature." Similar for calculated patterns at the bottom of Fig. 4A.

We have added descriptions of the P-T conditions for all calculated patterns.

Yakusheva et al. is cited as ref. 24, but as far as I can tell ref. 24 is not Yakusheva et al., nor is this reference in the bibliography.

We thank the author for noticing this error. We have corrected the references.

RESPONSE TO REVIEWER 2

I appreciate the efforts of Dattelbaum et al in addressing the concerns raised in the review of their submission. While many comments have been addressed, I remain concerned about the SAXS analysis.

While the q range has been expanded, this should place even greater rigour on the modelling of the data. The approach has changed from a straightforward spherical model (where fit parameters are directly related to structure e.g. radius of gyration, contrast etc.) to a Guinier-Porod model and background with, in some cases, a two-layer Guinier-Porod model.

Unquestionably, the models appear to fit the data but the table of fitting parameters and uncertainties remain of concern and do not add much structural information.

We agree with the reviewer's comment that the SAXS data provides limited structural information. However, due to the general inaccessibility of structural information pertaining to the reaction products at the time scales probed here we emphasize that any information, even limited to bounds on potential length scales, is highly valuable.

In fact, the Guinier-Porod model has been described as the model to use when one does not have an appropriate model.

Again, we agree with the reviewer's comment and our choice of this empirical model was motivated by the need to not over-interpret the SAXS data due to its limited information content.

The criticism of such an approach is that the extracted parameters are often not used to elucidate anything structural and are usually merely quoted as fitting parameters, as they are here. In principle, detailed information can be obtained from such a model by drilling down further but, given the uncertainty in them here, this would not be possible.

As the reviewer states, here it is not possible to extract detailed quantitative information from all of the model parameters. For example, we were unable to extract the number density or electron density contrast of the particles from the G values. This was due to the interdependence of these quantities on the parameter G and the inability to independently measure them under the dynamic conditions studied. However, the values of R_g obtained from the model do provide a shape-independent indication of the length scales associated with the product structures and in a manner that does not impose a particle shape (e.g. spherical) or morphology which would not be directly supported by the measured data. Further, the lower limit of the measured q space and the existence of a low- q power law dependence to the scattering provides an upper bound for larger length scales associated with the products. Due to the experimental limitations, we have restricted our interpretation of the SAXS results to describe only shape-independent length scales and lower bounds for length scales associated with the products.

For example, Table 1, the fitting uncertainty on G_2 is extremely large in all cases except for #237 which leads one to conclude that the fits are not particularly robust. Ditto Table S4 for G_1 .

We agree that the large uncertainty in G_2 is large due to the fact that the corresponding feature in the SAXS pattern is small compared to the dominant low- q power law and near the upper end of the measurement window in q . This does indeed shed doubt on the prominence of the associated length scale in the products. However, we would also like to clarify that the large uncertainty in the Guinier scaling parameters (G_1 or G_2) is also tied to the uncertainty in R_g since the intensity in the Guinier region is proportional to $G \cdot \exp(-q^2 \cdot R_g^2/3)$. In fact, the uncertainty in G obtained from our error analysis is almost directly correlated to the uncertainty in R_g . Since R_g is the structural parameter of interest, we find the robustness of the fits is best ascertained by examining the error in R_g where the errors correspond to a 20-40% uncertainty in the particle sizes. The text does point out that the scattering is dominated by the low- q power law and further provides the parameters and chi squared goodness of fit comparison for models with and without the inclusion of this Guinier-Porod level.

In my view, it is simply large objects that are generated whose size is outside the measurement window.

Again, we agree with the reviewer and have made efforts to further emphasize that the Guinier-Porod contribution corresponding to G2 is minor and the dominant contribution to the SAXS comes from objects larger than the measurement window and is manifested as a low-q power law. We have maintained the 2-level model description for completeness.

Other comment: Where is the fitted data for #294 in Figure 2? The parameters are listed for the fit but are not shown.

This was omitted to make the figure more easily digestible. The shape of the SAXS pattern and the quality of the fit are comparable to those displayed in the figure.

In addition, the authors state that they have employed time-resolved SAXS but there is no time evolution of data and associated fits illustrated as a function of time.

We thank the reviewer for this comment. Indeed the data presented here are for single snapshots in time and should not be described as ‘time-resolved’. We have removed this terminology except in reference to previous work.

In summary, as far as the SAXS component of the manuscript is concerned, while the experimenters should be congratulated for conducting such a complex experiment, I have strong doubts about the utility of the technique as applied here.

We acknowledge the limitations of the SAXS data collected here but emphasize that they do support the coexistence of a solid and fluid product mixture capable of generating electron density contrast and that they provide important constraints on the length scales associated with the products.

REVIEWERS' COMMENTS

Reviewer #1 (Remarks to the Author):

This substantial revision of the original manuscript sufficiently addresses my concerns, and I recommend publication of the paper in Nature Communications. In particular, the authors' consideration of variation of the 002 line (and related annotations in Fig. 4) support their case. Generally, I agree with the authors that carbon-containing systems can be very complex, and incisive observations (such as the 002 shift) can significantly contribute to our understanding of carbon phase transition mechanisms.

Reviewer #2 (Remarks to the Author):

In their revised submission Dattelbaum et al have essentially conceded all of the issues previously raised by this reviewer as far as SAXS analysis is concerned. It is difficult to provide any further comments except that, at best, the SAXS indicates that the products from run 292 have average dimensions that are partially within the observable range whereas those from the other runs are outside, placing an approximate lower limit on their size. It is not the case, as the authors state that, the data provide "an upper bound for larger length scales associated with the products" but rather a lower bound. Alas, after two reviews, the authors and I have probably reached the same conclusion that the SAXS data provide little additional information in this study and the major findings arise from the XRD data.

Manuscript ID: NCOMMS-20-27034B, "Carbon clusters formed from shocked benzene," D. M. Dattelbaum et al.

Response to Reviewers' Comments

Reviewer #1 (Remarks to the Author):

This substantial revision of the original manuscript sufficiently addresses my concerns, and I recommend publication of the paper in Nature Communications. In particular, the authors' consideration of variation of the 002 line (and related annotations in Fig. 4) support their case. Generally, I agree with the authors that carbon-containing systems can be very complex, and incisive observations (such as the 002 shift) can significantly contribute to our understanding of carbon phase transition mechanisms.

We thank the reviewer for his/her comments and critical review.

Reviewer #2 (Remarks to the Author):

In their revised submission Dattelbaum et al have essentially conceded all of the issues previously raised by this reviewer as far as SAXS analysis is concerned. It is difficult to provide any further comments except that, at best, the SAXS indicates that the products from run 292 have average dimensions that are partially within the observable range whereas those from the other runs are outside, placing an approximate lower limit on their size. It is not the case, as the authors state that, the data provide "an upper bound for larger length scales associated with the products" but rather a lower bound. Alas, after two reviews, the authors and I have probably reached the same conclusion that the SAXS data provide little additional information in this study and the major findings arise from the XRD data.

We thank the reviewer for his/her comments and critical review.